# Mechanisms of human dynamic object recognition revealed by sequential deep neural networks

**Lynn K. A. Sörensen**[1,2]\*, **Sander M. Bohté**[3,4,5], **Dorina de Jong**[6,7], **Heleen A. Slagter**[8,9], **H. Steven Scholte**[1,2]

**1** Department of Psychology, University of Amsterdam, Amsterdam, Netherlands, **2** Amsterdam Brain & Cognition (ABC), University of Amsterdam, Amsterdam, Netherlands, **3** Machine Learning Group, Centrum Wiskunde & Informatica, Amsterdam, Netherlands, **4** Swammerdam Institute of Life Sciences (SILS), University of Amsterdam, Amsterdam, Netherlands, **5** Bernoulli Institute, Rijksuniversiteit Groningen, Groningen, Netherlands, **6** Istituto Italiano di Tecnologia, Center for Translational Neurophysiology of Speech and Communication (CTNSC), Ferrara, Italy, **7** Università di Ferrara, Dipartimento di Scienze Biomediche e Chirurgico Specialistiche, Ferrara, Italy, **8** Department of Experimental and Applied Psychology, Vrije Universiteit Amsterdam, Amsterdam, Netherlands, **9** Institute of Brain and Behaviour Amsterdam, Vrije Universiteit Amsterdam, Amsterdam, Netherlands

\* lynn.soerensen@gmail.com

**Data Availability Statement:** All results and code to reproduce these results can be accessed on the Open Science Framework (https://osf.io/c9gs8/).

## Abstract

Humans can quickly recognize objects in a dynamically changing world. This ability is showcased by the fact that observers succeed at recognizing objects in rapidly changing image sequences, at up to 13 ms/image. To date, the mechanisms that govern dynamic object recognition remain poorly understood. Here, we developed deep learning models for dynamic recognition and compared different computational mechanisms, contrasting feedforward and recurrent, single-image and sequential processing as well as different forms of adaptation. We found that only models that integrate images sequentially via lateral recurrence mirrored human performance (N = 36) and were predictive of trial-by-trial responses across image durations (13-80 ms/image). Importantly, models with sequential lateral-recurrent integration also captured how human performance changes as a function of image presentation durations, with models processing images for a few time steps capturing human object recognition at shorter presentation durations and models processing images for more time steps capturing human object recognition at longer presentation durations. Furthermore, augmenting such a recurrent model with adaptation markedly improved dynamic recognition performance and accelerated its representational dynamics, thereby predicting human trial-by-trial responses using fewer processing resources. Together, these findings provide new insights into the mechanisms rendering object recognition so fast and effective in a dynamic visual world.

## Author summary

Our visual world is both stable and dynamic: even within a single glance, a scene may change dramatically. Brains thus need to balance integration of information over time to

**Funding:** This work was funded by a Research Talent Grant (406.17.554) from the Dutch Research Council (NWO, https://www.nwo.nl/) awarded to HSS, LKAS, SMB and HAS. The funder had no role in study design, data collection and analysis, decision to publish, or preparation of the manuscript.

**Competing interests:** The authors have declared that no competing interests exist.

create stable percepts with sensitivity to changes in sensory input, e.g., to rapidly recognize new objects. How do brains and, in particular, visual systems achieve this? Here, we addressed this question by having humans and different neural network models perform the same object recognition task in which sequences of images were shown in rapid or slow succession. We observed that models treating images as a continuous sequence by integrating its processing over time reproduced human performance patterns better than models processing every single image at a time. Furthermore, models equipped with sensory adaptation, a form of stimulus habituation, better recognized objects in faster sequences and more efficiently captured human behaviour. These findings show that lateral recurrence and adaptation jointly enable object recognition across a wide variety of time scales, suggesting a critical role for these mechanisms in dynamic vision.

## Introduction

Our visual world is dynamic, and even within a single glance, or fixation, a scene may change dramatically, for example, from one movie frame to another, in traffic or when we turn our heads. Object recognition in primates is mostly robust towards such rapid changes and can operate over a wide range of time scales. One visual task that challenges our visual system to the limits of said ability is rapid serial visual presentation (RSVP [1]), where participants have to detect a target image in a stream of rapidly presented real-world scene images. Remarkably, human observers are still capable of detecting a target image even if images are presented for as briefly as 13 ms per image [2–4]. Performance further increases if images are shown for longer periods of time. While primate object recognition is increasingly well explained by modern deep learning models optimized for computer vision, these frameworks still fall short in explaining object recognition across varying timescales, as well as accounting for a dynamic stimulation (i.e., a constant stream of visual change). Here, we take steps towards modelling human dynamic object recognition within a glance by investigating the functional contributions of different neurally plausible computational mechanisms and linking them back to the behavioural patterns of human observers.

A key debate in the literature on object recognition centres on the question to what degree (core) object recognition [5,6] relies on recurrent processing. In line with feedforward models of perception, one key proposal is that object recognition based on only 13 ms per image can be achieved solely through feedforward processing [2]. That is, the first wave of afferent neural activation through the ventral stream is thought to carry sufficient information to enable core object recognition [5,7,8] for every image in the stream. In this view, recurrent processing within and between areas in the visual hierarchy does not substantially contribute to performance, as the subsequent image in the stream [2,9–12] acts as a mask, preventing recurrent processing of the preceding image [13]. In contrast, longer presentation durations are thought to give increasing opportunity for recurrent processing, resulting in more accurate object recognition. The notion that feedforward processing suffices for visual comprehension in such dynamic situations is however questioned by a set of more recent studies. In one RSVP study, for example, the typical natural scene distractor images in the image streams were replaced by different kinds of masks (i.e., images with geometric patterns and textures), which more effectively interfered with recurrent processing. It was found that using such masks, performance is no longer above chance for images presented for just 13 ms [4], suggesting that recurrent processing is in general critical for object recognition within a glance. In line with this notion, recent MEG and EEG studies show that the visual cortex can maintain multiple concurrent

image representations during RSVP streams [14–16], even at durations of up to 17 ms/image [14], a process that is likely facilitated via local and top-down recurrence. Together, these findings suggest that object recognition during dynamic visual stimulation, such as ultra-rapid RSVP streams, may not be achieved based on feedforward processing alone, but in part also relies on recurrent processing. What is still unclear is to what degree such recurrent processing is functionally necessary for rapid object recognition performance.

Next to re-entrant processing, neural adaptation may also be key to dynamic object perception within a glance. Performance on an RSVP task not only requires the extraction of image features relevant for object recognition, but also relies on the segmentation of an image stream into different visual events. The usefulness of segmentation becomes apparent when considering a system capable of performing at both fast and slow presentation durations: for long presentation durations, feature extraction relies on the accumulation of information over time, for instance, via recurrent processing. For short presentation durations, however, it would not be helpful to keep integrating image features over time, since information across images would be merged. How can a visual system succeed at dynamic object recognition both at short and long time-scales? Rapid sensory adaptation may be a strategy to achieve this [17,18], as it efficiently decorrelates temporal inputs [19–22]. We here refer to rapid sensory adaptation as suppression of a neuron's activity based on its recent sensory history [23]. While not commonly associated with performance on an RSVP task, neural adaptation is a central component in modelling neural integration across temporal contexts (e.g., [24]) and has been proposed to serve as a key means to reduce the saliency of recently seen stimuli in sensory processing [23,25–27]. A recent study by Vinken and colleagues [28] showcases how the implementation of stimulus-specific adaptation in a feedforward deep convolutional neural network (DCNN) can recover a wide range of neural and behavioural properties, including the neural signatures of stimulus deviancy detection along the visual ventral hierarchy. Adaptation is thus conceivably also crucial for object perception in dynamically changing environments. Specifically, adaptation may support performance during dynamic object recognition by increasing the salience of a new image in a sequence, particularly at short time scales.

Here, we examined the role of neural recurrence and adaptation in dynamic object recognition using deep convolutional neural networks (DCNNs). Recent years have seen a surge in using DCNNs to account for the neural, behavioural and architectural properties associated with object recognition and the visual ventral stream in primates [29,30]. While feedforward architectures are still widely used in this endeavour [31], recurrent DCNNs may be superior in describing the computational principles underlying object recognition given their architectural realism [32], behavioural dynamics [33] and ability to fit neural data [34–36]. One key advantage of recurrent neural networks over their feedforward and parameter-matched counterparts is that due to their recurrent processing, they are naturally endowed with a notion of time, a key parallel to biological vision. Yet, while recurrent DCNN have become more accurate at predicting the neural and behavioural dynamics in response to a single static image, an open question is how temporal processing in recurrent models can be leveraged to understand the processing of dynamically changing visual inputs such as during RSVP. After all, object recognition is typically achieved in the context of a stream of dynamic inputs to the brain. Adopting such a constraint may help to identify more predictive models of how the brain solves object recognition across different time scales.

We here addressed this gap in knowledge by transforming a recurrent DCNN, *BLnet* [33], optimized for single-image recognition, into a model for sequential object recognition, *BLnext*. We chose *BLnet* as our starting point instead of other recurrent DCNNs (e.g., [35]) because *BLnet* is designed to receive sequences as its input. By applying some modifications to *BLnet* (e.g., a change to the batch normalization statistics), we obtained a model (*BLnext)* suitable for

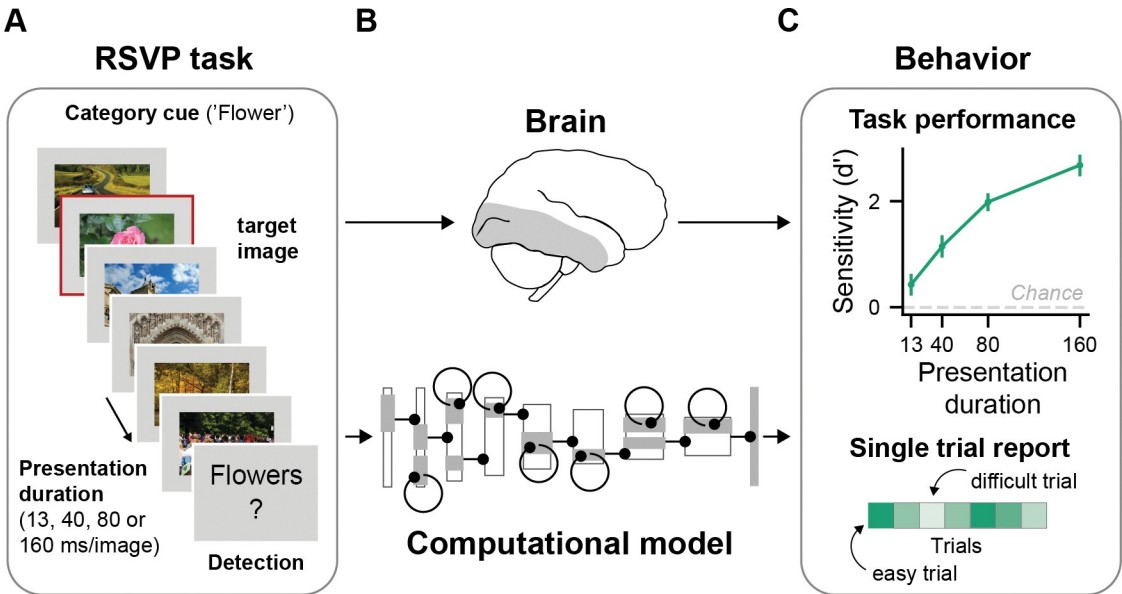

**Fig 1. Our computational approach towards understanding dynamic object recognition within a glance. A** Schematic of the RSVP task presented to both human participants and computational models. Participants and models were tasked to detect whether an image belonging to pre-specified target category was present in an image sequence. Every trial contained a unique target category and always consisted of new images. Data was collected for multiple presentation durations for the same trial sequences. Original stimuli were replaced with comparable license-free images. **B** To understand the computational mechanisms that shape neural computations when dealing with rapidly changing sensory streams, we build a variety of computational models aiming to capture this mapping. **C** Human behaviour serves as the explanatory goal for the computational models. To this end, we tested all computational models for their ability to perform on the same RSVP task as humans and to capture human single-trial reports. The upper panel shows the task performance, that is, the mean perceptual sensitivity across 36 participants, as a function of image presentation duration (13, 40, 80 and 160 ms). Error bars show the 95% confidence interval for the mean sensitivity. Participants performed above chance at all presentation durations (all $p's < .0001$). The lower panel illustrates that trials varied in difficulty. A good model should capture mean performance levels across presentation durations (upper panel) as well as allow one to predict which task trials should be easy or hard given a certain presentation duration (lower panel).

sequential object recognition. Using *BLnext* we could systematically investigate the computational mechanisms critical for object recognition in dynamically changing visual environments. Specifically, we studied the added cost of sequential recognition by comparing it to the simpler task of single-image recognition (i.e., processing a single image at a time), and ask how mechanisms such as lateral recurrence and sensory adaptation may support recognition in such a dynamic setting. In the next step, we linked these mechanisms to human behaviour and determined how the performance of different models, serving as our hypotheses, compared to that of humans on the same RSVP task (see Fig 1). In particular, we compared the performance of feedforward models with recurrent ones, the performance of models that process images in isolation with those that process sequential inputs, and the performance of models with sensory adaptation with those without. Finally, we assessed whether a single model could predict human representational dynamics on this RSVP task by predicting the single-trial behaviour across all presentation rates.

We find that recognizing objects in a dynamic stream of inputs is indeed a challenging task (compared to single-image recognition), yet that sequential recognition performance is critically improved by both lateral recurrence and sensory adaptation. Comparing different models to human behaviour, we observed that recurrent models with sequential image processing (*BLnext*) selectively capture the gradual increase in perceptual sensitivity with longer stimulus presentation durations displayed by humans on the same RSVP task. These models with multiple processing steps also showcase that it is possible to perform an RSVP task based on

feedforward processing alone (a single model step), and furthermore, that local recurrent processing provides a plausible account for how increased stimulus duration facilitates target detection. In predicting single-trial human report rates, we observed that only *BLnext* models showed a match between presentation duration and model processing steps (equivalent to processing over time), with models with fewer steps providing the best explanation for human behaviour at the fastest durations and vice versa. While other (non-sequential) recurrent networks overall featured a similar explanatory power, they did not capture this temporal correspondence. Intriguingly, adaptation was not only essential for boosting task performance across all studied dynamic recognition tasks, but also substantially accelerated a model's representational capacity enabling it to account for single-trial human behaviour using fewer processing resources (i.e., fewer model steps). Taken together, we find evidence that lateral recurrence and adaptation are jointly effective at enabling dynamic object recognition within a glance.

## Results

What kind of computational mechanisms enable dynamic object recognition within a glance in human observers? To address this question, we developed a variety of computational models, serving as our hypotheses and tested their performance on two different dynamic recognition tasks (standard sequential object recognition and a cued RSVP task). For the RSVP task, we compared model performance with those of human observers, and assessed how well different models predicted single-trial human participant data performing the same task.

### Method's summary

We here study the computational mechanisms underlying human sequential object recognition. During sequential object recognition, a model receives a video stream as an input consisting of multiple images. The processing of preceding images can thus influence the processing of current and future images. In this mode of recognition, no additional information on the timing or number of target images is provided. This contrasts with standard object recognition tasks that we refer to as single-image recognition. During single-image recognition, a model is presented with a single image at a time and artificially reset before processing another image. In single-image models, the processing of one image to the next is completely separated. If thought of as a sequential process, a single-image model would also receive external information about the timing of potential targets (such that the model is reset for processing a new image). Compared to biological visual processing, we therefore consider this an easier and more artificial form of object recognition. While less biologically plausible, single-image models have been very successful in explaining core object recognition in brain and behaviour [37], and thus served us here as useful points of reference.

The sequential visual processing performed by participants during an RSVP task is likely dramatically different from the processes captured by single-image models, since such models do not have to deal with the transition between images. To test the importance of time and sensory memory for accounting for human sequential object recognition, we therefore developed another model class, *BLnext*, capable of processing input sequences of varying duration (sequential models), where the processing of past stimuli influences the processing of the current stimulus.

*BLnext* consists of bottom-up feedforward and lateral recurrent weights (see Fig 2A) and inherits both its architecture and weights from *BLnet* [33], a recurrent object recognition model optimized on a large-scale image database (Ecoset, [38]). *BLnet*, however, is only optimized for eight recurrent model steps. Since we wanted to evaluate considerably longer sequences, we developed a version in which *BLnext* was able to operate sustainably over longer sequences by adjusting its batch normalization regime (see *Methods–Computational models*

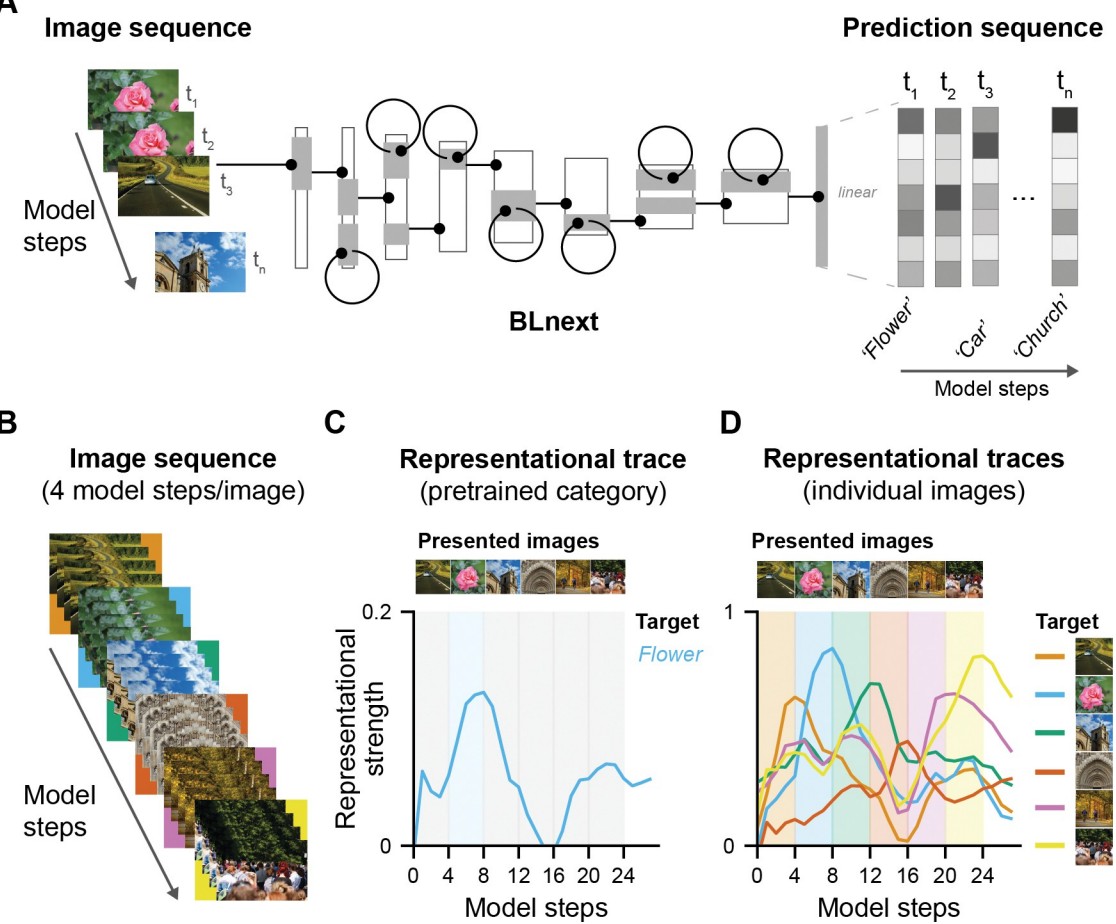

**Fig 2. BLnext, a model for sequential object recognition. A** *BLnext* accepts image sequences as inputs. These sequences are processed with bottom-up feedforward and lateral recurrent weights, resulting in a prediction sequence as output. The first step in the prediction sequence corresponds to a single feedforward pass induced by the first sample of the input sequence. For later prediction sequence steps, the lateral recurrent weights also contribute to the prediction. This way, preceding time steps can influence the processing of the current time step, either adding to the accumulated evidence if the same image is processed, or moving the model into a new direction for a new image [33]. Model illustration is adapted from [33]. **B** To model different presentation durations, we varied the number of repetitions per image in a sequence, shown here are 6 images, presented for 4 model steps per image each. **C** For pretrained categories, representational strength can be computed for every model and presented image for a given target category presented in the image sequence (flower) as the correlation with the categorical one-hot vector. Representational strength of this target category (flower) was highest for model steps during which the corresponding target was presented (blue shaded area). **D** Alternatively, a model's predictions can also be interpreted with regard to its similarity to a single-image-level representation, that is, the representational strength of that image. An image-level representation defined by the output state of a model presenting a given target image in isolation. In this example, the representational strength of the six presented images is displayed. The shaded coloured areas indicate the respective presentation duration of these images (4 model steps/image) and their location in the image sequence. For all six images, representational strength peaked during image presentation, gradually increasing as a function of presentation duration, highlighting the ability of *BLnext* to capture sequential image processing.

for more information). After processing an image through its hierarchy, *BLnext* reflects its current state in its output layer (Fig 2A), where a single model step in this output layer corresponds to a full feedforward sweep in the model. Due to its lateral recurrence, with more and more model steps, preceding model steps increasingly impact the processing of the current model step, thereby accumulating evidence for as long as a given image is presented.

With such a sequential model, we can achieve two key goals: firstly, we can use this model as a testbed to understand the functional contributions of different neurally plausible mechanisms, such as lateral recurrence and sensory adaptation to sequential object recognition.

Secondly, we can link these models to a broader range of human behaviours by mimicking different presentation durations used in human RSVP experiments, including those used in our experiment in human participants (i.e., 13, 40, 80 and 160 ms/image).

To achieve these goals, we presented *BLnext* with sequences consisting of a varying number of samples, or model steps per image (e.g., Fig 2B shows a sequence with 4 model steps per image). To evaluate how different mechanisms might support sequential recognition, we tracked the top 5 most activated output nodes for every model step (see Fig 2C for an example of a pretrained category), thereby obtaining a measure of task performance as a function of presentation duration for the different models. For our second goal, understanding the model's link to human performance, we evaluated the model's responses to the same experimental trials as performed by human participants and recorded the representational strength linked to a given target category or image (see Fig 2D for an example and *Methods* for more details). This was necessary since the human RSVP experiment contained many images and target categories that were not well captured by the pretrained categories. In particular, we computed the representational distance between *BLnext*'s linear output states during the RSVP stream and another target representation (*representational strength*, Fig 2D). Traditionally, as used for testing performance, this target representation is a categorical one-hot vector, indicating the match with a pretrained category. Since the pretrained categories did not match, we adopted an alternative way to track the representations of either a category prototype or a single image (independent of its category). To do this, we used the final output state of *BLnet* representing either a category prototype (i.e., mean representation across a group of images) or a single image as a target vector (Fig 2D). This choice followed the logic that representations evoked by an image in *BLnet* are the best possible representation of a given image given this architecture and these trained weights (which are identical in *BLnext*). Importantly, using this approach, we could track how representations of individual images or categorical prototypes waxed and waned throughout an image sequence (see Fig 2D for examples of image traces).

To understand how performance on a sequential recognition task compares to that on a single-image recognition task using the same stimuli (yet without the added difficulty of sequential processing), we compared the sequential recurrent model *BLnext* with a range of single-image reference models, in particular its single-image recurrent counterpart, *BLnet*, as well as with two single-image feedforward architectures: *B*, which has the same number of parameters as a single feedforward pass (thus a single model step) as the recurrent models (*BLnet* and *BLnext*), and *B-D*, which has the same number of parameters as *BLnet* and *BLnext* but without its recurrent connectivity. All models were trained on the same dataset [38], and were matched in architecture [33] (see Table 1 for an overview).

## Sensory adaptation and lateral recurrence jointly improve sequential object recognition performance

In our first analysis, we evaluated how two main, neurally plausible mechanisms—lateral recurrence and sensory adaptation—may contribute to sequential object recognition. In contrast to performance on a standard object recognition task (using single images), we here focus on the added difficulty of recognizing target images presented in rapid succession, akin to the inputs received by biological visual systems (see Fig 3A for a schematic). In particular, we investigated how the two mechanisms shape the ability of a model to recognize targets presented sequentially across varying presentation durations (i.e., model steps/image). For lateral recurrence, we hypothesized that slower sequences (i.e., more steps/image) should enable the network to integrate its evidence more effectively, enhancing recognition performance. Moreover, neural adaptation may also be useful during sequential object recognition, since it has

**Table 1. Overview of all sequential and reference models.**

| model | recognition type | lateral recurrence | adaptation | parameters |
|---|---|---|---|---|
| *BLnext* | Sequential | yes | none | 28.9 million |
| *BLnext (exp.)* | Sequential | yes | exponential | 28.9 million |
| *BLnext (pow.)* | Sequential | yes | power-law | 28.9 million |
| *BLnet* | image-by-image | yes | none | 28.9 million |
| *BLnet (exp.)* | image-by-image | yes | exponential | 28.9 million |
| *BLnet (pow.)* | image-by-image | yes | power-law | 28.9 million |
| *B* | image-by-image | no | none | 11.0 million |
| *B-D* | image-by-image | no | none | 28.9 million |

been proposed to reduce the saliency of previously seen images thereby potentially leading to a better discrimination across images and hence to improved recognition performance. To test these predictions, we implemented three different sequential model variants and evaluated their recognition performance on sequentially presented images from the Ecoset test set.

Our implementation of neural adaptation closely followed Vinken and colleagues (2020) [28], called activation-based intrinsic suppression (Fig 3B). With this approach, suppression is built up as a function of a unit's past activation, and this suppression is subsequently subtracted from new incoming activation prior to applying the rectifying activation function (see Fig 3B for an illustration). Here, we used the same approach developed for a feedforward DCNN [28] in a recurrent DCNN (Fig 3B, *exponential adaptation*). While exponential functions are often adopted to model neural adaptation at fast time scales [23,39] and are attractive for their simplicity, they have been criticized for only capturing limited time scales of adaptation [18,40]. Since the questions addressed here involved quite different timescales (13–80 ms/image), we also implemented a network with a power-law adaptation mechanism using a sum of exponentials [18,40–42], another common form of neural adaptation observed experimentally [18,23,41,43–45] (see *Methods* for more details). Fig 3B highlights some of the differences between an exponential and power-law adaptation mechanism on a single-unit level: While both types of adaptation result in very comparable activation reductions over short time scales ($r_{exp}$, $r_{pow}$), for later model steps, there are increasing differences between these approaches, with a much less pronounced suppressive effect over time for power-law adaptation.

Neural adaptation may also interact with recurrent processing. In contrast to feedforward DCNNs as used before [28], recurrent DCNNs undergo qualitatively different representational stages over time–every recurrent computational step adds to the information of earlier model steps, resulting in a gradual increase in information and consequently, performance (e.g., [33]). As neural adaptation also affects stimulus processing over time, neural adaptation and recurrency may have interactive effects. The example representational traces in Fig 3C illustrate this: Comparing the model's representational trace for the same trial across the three different *BLnext* variants highlights some qualitative differences in dynamics between models with and without adaptation. While a network without adaptation maintains a representational trace of the target image for as long as it is presented, in a network with adaptation, this representational trace starts to decay after its peak, even while the image is still shown (see Fig 3C). Note that all networks reached a comparable representational strength, but the networks with adaptation reached this peak slightly faster and their image traces decayed earlier.

To assess the wider functional contributions of lateral recurrence and adaptation to dynamic object recognition, we devised a sequential recognition task in which we presented the different models with image sequences of varying duration. Importantly, we used the test set of Ecoset [38] to construct the trials, making sure that all models had the relevant training

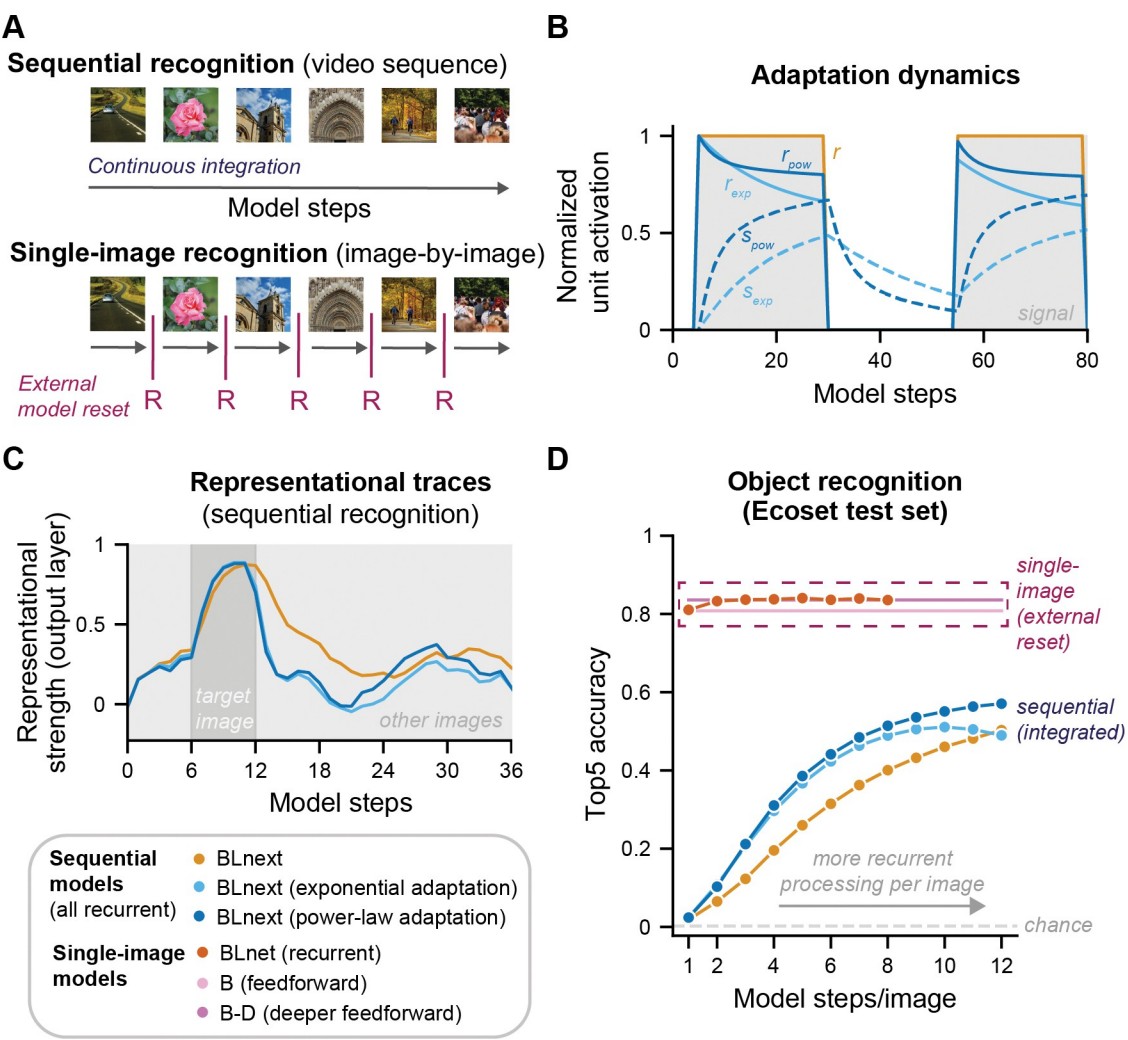

**Fig 3. Augmenting *BLnext* with different forms of adaptation boosts sequential object recognition. A** We here study the challenging task of sequential object recognition, during which a model simply receives a video stream, akin to the inputs received by biological visual systems–without any indication of the timing and number of presented images (top panel). We compare this to the easier task of single-image recognition, during which recognition is tested on a single image at a time and is artificially reset for every new image (lower panel). If thought of as a sequence, this type of recognition presupposes knowledge of the exact timing of every image change. Single-image models serve here as reference models that capture stimulus difficulty without the added difficulty of sequential processing. To understand the potential contributions of lateral recurrence and adaptation to sequential object recognition, we compared three different candidate models: A lateral recurrent model without adaptation (*BLnext*, orange) evaluated across a different number of recurrent steps (incl. no recurrent processing for a single model step), the same model with exponential adaptation (*BLnext–exponential*, light blue) and one with power-law adaptation (*BLnext–power-law*, dark blue). **B** Suppression and activation evolve over time in a network unit with adaptation. As can be expected, there is no reduction in activation without adaptation ($r$, orange) over time, while for both exponential and power-law adaptation (light and dark blue, respectively), an ongoing incoming signal (shaded grey area) leads to a build-up in suppression ($s_{exp}$, $s_{pow}$), resulting in a reduced activation ($r_{exp}$, $r_{pow}$). Without a signal, these suppression indices decay and can potentially result in an offset at the arrival of a new signal. For clarity, this example does not show any recurrent interactions. **C** Evaluating a trial example (same as shown in Fig 2B–2D) for sequential recognition highlights that in contrast to a recurrent model without adaptation, both forms of adaptation are associated with suppressed representations of the target right after the target image is presented. **D** Lateral recurrent models and in particularly those with adaptation improve in their sequential object recognition performance on the Ecoset test set (500 image sequences based on 6 random images) on longer sequences. Top5 accuracy was chosen to account for lingering representations of previously processed images during sequential processing. As a reference, performance on the easier task of single-image recognition (using the same images) is shown (dashed purple frame). Single-image recognition performance reflects the difficulty of the randomly chosen images (without any interference due to sequential processing).

and output categories (565 possible categories) to successfully recognize the presented images. We evaluated all sequential models on 500 randomly generated image sequences, each consisting of 6 images (all from a different category) and thus 6 targets. We showed these sequences repeatedly to the models while varying the number of model steps per image and tested their performance.

First, we observed that sequential recognition performance was only slightly above chance for the fastest sequences (i.e., one model step/image, see Fig 3D) but improved for slower sequences. This suggests that feedforward processing alone might not be sufficient for sequential object recognition on this task. This contrasts with the performance on the easier single-image task, indicating that models processing every image in isolation display high performance even after a single model step. This points to the added difficulty introduced by sequential recognition. Only when images were presented at a slower pace did the model succeed at this task, displaying a steady increase in Top5 accuracy with increasing image durations (Fig 3D). This likely reflects increasing contributions from lateral recurrence, as we proposed. Second, we found that adaptation had a clear effect on recognition performance: starting already for fast sequences (with only 2 model steps/image), both forms of adaptation reached higher Top5 accuracy than the model without adaptation. For longer sequences, particularly a power-law-based mechanism proved effective at boosting recognition in comparison to a plain sequential model. This finding raises the question whether the performance benefits derived from adaptation mechanisms are specific to sequential object recognition or may also be observed in recurrent models processing one image at a time. S2 Fig clarifies that the latter is not the case. This suggests that the combination of lateral recurrence and sensory adaptation is particularly beneficial for the challenging task of sequential object recognition.

To situate the performance of sequential models it is informative to compare them to those of single-image model: If a sequential model achieved perfect segmentation between images in a sequence, performance should be comparable to that of a model performing single-image recognition (incl. external model resets, see Fig 3A), thus only influenced by the image features. This makes that performance of single-image models can be regarded as an estimate of the upper performance limit for a given stimulus set. We found that for all examined models, there was a performance cost incurred by processing images in a sequential fashion, as they exhibited slightly poorer object recognition even on the slowest sequence trials (i.e., >10 model steps/image) compared to any model performing single-image recognition (see Fig 3D).

Dynamic object recognition is a challenging task because it does not only require solving what can be seen (i.e., single-image recognition) but also requires resolving the transitions between images (i.e., the interference of lingering representations). Taken together, our findings provide direct evidence that lateral recurrence and sensory adaptation critically contribute to solving this challenge by showing improved recognition performance in models with more opportunity for lateral recurrence and in models using different forms of sensory adaptation. Crucially, our results highlight that the interaction between those mechanisms is beneficial for sequential recognition.

## Sequential, lateral recurrent image processing is key to recovering human RSVP performance levels

Having established the functional benefits of lateral recurrence and adaptation for sequential object recognition, our next goal was to compare different models with regard to their ability to perform on an RSVP task, previously studied in humans. This stands in the tradition of using task-optimized models as mechanistic models of neural processing and behaviour [34,37,46].

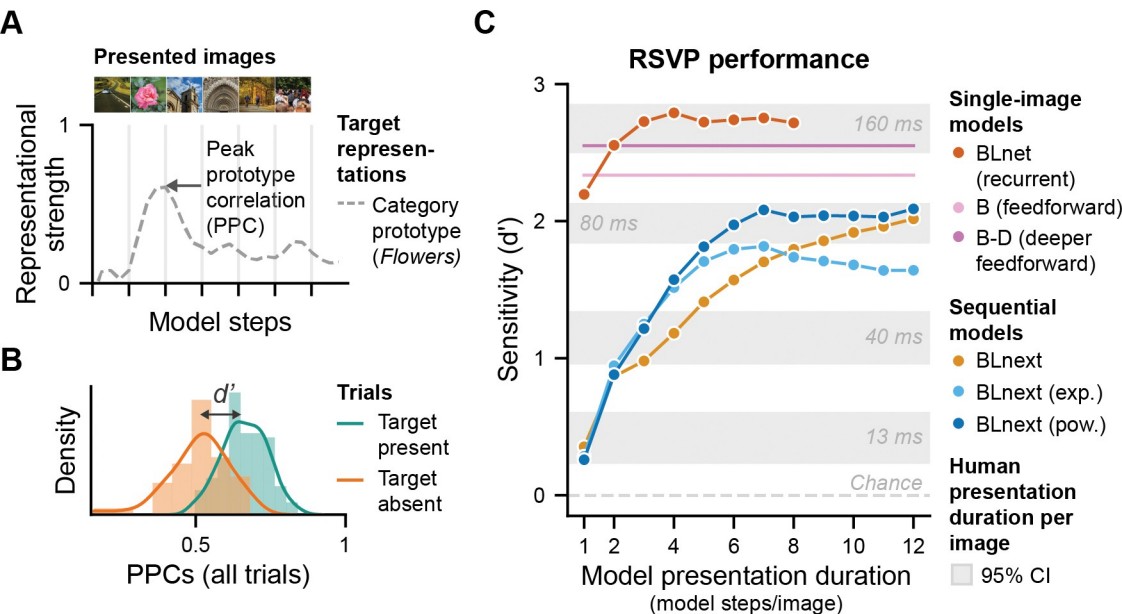

**Fig 4. Comparing perceptual sensitivity across models and to human observers. A** Peak prototype correlation (PPC), a proxy for sensory evidence for a given category (here flowers), fluctuates across model steps in a given trial in the *BLnext* output traces. Here only the PPC for a single category is shown across an image sequence. **B** Perceptual sensitivity (d') was obtained by comparing the distributions of the PPC values for trials with and without a target for a given model. **C** This panel shows performance on the RSVP task for the different models and humans. Single-image models vastly outperform human observers at all presentation durations, except for 160 ms. In contrast, sequential models display a gradual increase in d' as model steps per image increase, thereby mimicking the gradual increase in perceptual sensitivity shown with increases in presentation duration in human observers (see Fig 1C). Coloured dots represent model steps for models with recurrence.

To enable a comparison between models and humans, we translated the model's representational traces (sequential models) or the model's response to single images (single-image models) into a perceptual sensitivity measure summarizing performance across trials. Furthermore, human participants responded to a unique, and rather specific category cue (e.g., grassy hills) on every trial. To recreate a comparable situation for the models, we developed prototypes that served as the target representation on a given trial. In brief, trial-specific prototypes were based on a large-scale online image query specific to every trial category cue. Prototypes were then defined as the geometric centre of relevant category examples in a model's output space (see *Methods–Prototypes* for more details). Using these prototypes, we obtained representational traces specific to the trial's category cue and summarized a given trial based on the peak of this prototype correlation (*PPC*, see Fig 4A). Using these *PPC* values as a proxy for sensory evidence across trials, we estimated how well trials with and without a target were distinguished by the model (Fig 4B). This separation between the *PPC* distributions formed by trials with and without a target in turn corresponded to *d'*, a measure for perceptual sensitivity.

Comparing the different models (Table 1) based on their task performance revealed three interesting observations (Fig 4C). Firstly, it is evident that the single-image models, thus those that do not model RSVP streams as continuous but instead treat every image as a separate process, not only vastly outperformed all other models, but also human observers (for presentation durations shorter than 160 ms/image). This result means that this group of models, irrespective of being feedforward or recurrent, was too good compared to human observers (Fig 4C, *single-image models*). Even in a recurrent model, *BLnet* which showed an increase in performance with more model steps/image, already a single model step was associated with superior performance compared to human performance at 80ms/image. Taken together, this

indicates that modelling image processing as isolated events (i.e., single-image recognition) does not mimic human performance and that rather, models are needed that also address the sequential nature of input processing.

A second observation that can be made comparing the different models is that, in contrast to single-image models, models with sequential recurrent processing (*BLnext*) showed a gradual increase in perceptual sensitivity with an increase in model steps per image (Fig 4C, sequential models). These models also displayed comparable performance levels as reported for human observers for 13, 40 and 80 ms per image, respectively. They did not reach performance levels for a presentation duration of 160 ms per image (again, in contrast to the single-image models). These observations indicate that processing images as a sequence is important for capturing how the challenging nature of this task affects human neural processing up to 80 ms per image. That is, sequential object recognition is a much harder problem, not only requiring the ability to recognize objects, but also to dissociate them from each other in the dynamic input.

Having established that these sequential models feature similar performance levels as human observers, we can also use them to revisit some debates in the experimental literature. As mentioned in the introduction, based on current experimental results it was unclear whether object recognition in an RSVP is possible based on only feedforward processing in a recurrent network as suggested before [2]. Our sequential model highlights that this may indeed be possible, as indicated by the above chance performance after a single model step. While a single model step is most likely not identical to the neural feedforward sweep, our model can still be seen as a proof-of-principle that a locally recurrent architecture may display such abilities.

A third noteworthy finding of our model comparison is that within the sequential model class, models with adaptation achieved better perceptual sensitivities based on shorter model presentation durations (fewer model steps per image). This means that adaptation boosted task performance and helped to deal with dynamic inputs for medium-to-fast presentation durations (Fig 4C, blue vs. yellow lines). This finding supports the idea of a functional role for neural adaptation during dynamic visual processing and replicates the findings from Fig 3 using another set of images and behavioural measure. Of further note, only a *BLnext* model with power-law adaptation also maintained high performance levels with slower paced sequences, whereas an exponential adaptation model's performance suffered with more model steps per image. This finding directly connects back to past modelling work suggesting that power-law adaptation is better able to capture a wide range of timescales [18,40,41,44,45]. The performance benefits gained from combining *BLnext* with a neural adaptation mechanism also point to a larger computational principle in which recurrent computation may be rendered more efficient by being made susceptible to repetition, such as is achieved here with neural adaptation. Put differently, for representing a dynamic visual world, it may be adaptive to represent an input for as long as new information arrives, and to move on when information becomes redundant. Such redundancy can be effectively tracked with a neural adaptation mechanism. Our data suggests that combining (local) recurrence and neural adaptation may provide a neural system with just that trade-off between integration over time and sensitivity to change.

To summarize, evaluating a variety of computational models on their performance on an RSVP task, we first found that only models processing RSVP streams as sequences and with local recurrence well captured human performance up to 80 ms/image. Moreover, these *BLnext* models displayed above-chance performance at 13ms/image with just one model step, supporting a feedforward-only account of object recognition. In addition, recurrent processing brought about an increase in perceptual sensitivity for slower presentation durations, as also

seen in humans, highlighting the additional importance of recurrency in perception. Finally, our results demonstrate the functional benefits of neural adaption, and in particular power-law adaptation, for dealing with dynamic visual inputs.

## Only sequential, lateral recurrent models are predictive of human single-trial dynamics

Based on the previous results, we have identified a set of candidate models that recapitulate average human performance levels on an RSVP task. We next examined how well these models account for other properties of RSVP behaviour. For one, a good model of human dynamic visual perception should also explain variance in performance on a trial-by-trial basis. That is, the same trials should be easy and difficult to recognize for both models and humans. More-over, there should not only be correspondence in the trial-by-trial difficulty (mean sensitivities across trials, Fig 4C), but there should also be overlap in how trial performance changes across different presentation durations. For instance, some trials may always lead to the report of a target irrespective of the presentation duration, whereas on other trials targets are only detected with longer presentation durations. This overlap in variance between model and human reports across presentation durations can be used to further quantify which of our computational models best mimics human RSVP processing.

To this end, we compared the average report rate (for a given presentation duration) with a model's PPC (for a given number of model steps) on a trial-by-trial basis (see Fig 5A for examples). For target trials, a high report rate and high PPC means that both humans and models were likely to report the target (see Fig 5A upper left panel); for non-target trials, a low report rate and low PPC means that they were likely to report the absence of a target (see Fig 5A lower left panel). In both of these cases, model and humans agree in their assessment. Natu-rally, model and human single-trial reports could also disagree (see right panel in Fig 5A for two examples). Critically, in this comparison, all the specifics of the trial (i.e., the target cue and all other images in that sequence) are identical in the humans and model data. This allowed us to specifically test the alignment in perceptual difficulty between models and human observers. To summarize across trials, we used a Spearman correlation (see Fig 5B), where a high positive correlation reflects that trials with a high average report rate in humans corresponded to high PPC values in the models, and vice versa. Using this approach, we assessed the trial-by-trial correlations of all models (for different presentation durations, that is, model steps per image) for all human presentation durations (see Fig 5C). We here focused on presentation durations of up to 80ms/image corresponding to the maximum task perfor-mance achieved by the sequential models (Fig 4C).

Inspection of the trial-by-trial correlation profiles for different model and human presenta-tion durations (Figs 5C and S3) revealed some qualitative differences between the models. First, while all models reached high relative correlation values (compared to the noise ceiling, the highest possible correlation given the participants' variability) at 13 ms/image, recurrent models in particular reach or surpass the lower noise ceiling with at least one of their presenta-tion durations (number of model steps per image). These findings indicate that the weights from *BLnet*, a single-image model, can fully explain variance in the participant's single trial behaviour at 13ms/image. This is not achieved for any other presentation duration (40 and 80ms/image) and indicates that *BLnet* can less well capture human performance variability at these longer presentation durations. Furthermore, within the sequential recurrent models, we can discern some differences with regard to the model presentation duration reaching the lower noise ceiling at 13ms/image. In particular, models with adaptation show a sharp increase in predictivity that is specific to presenting an image for two model steps and that declines for

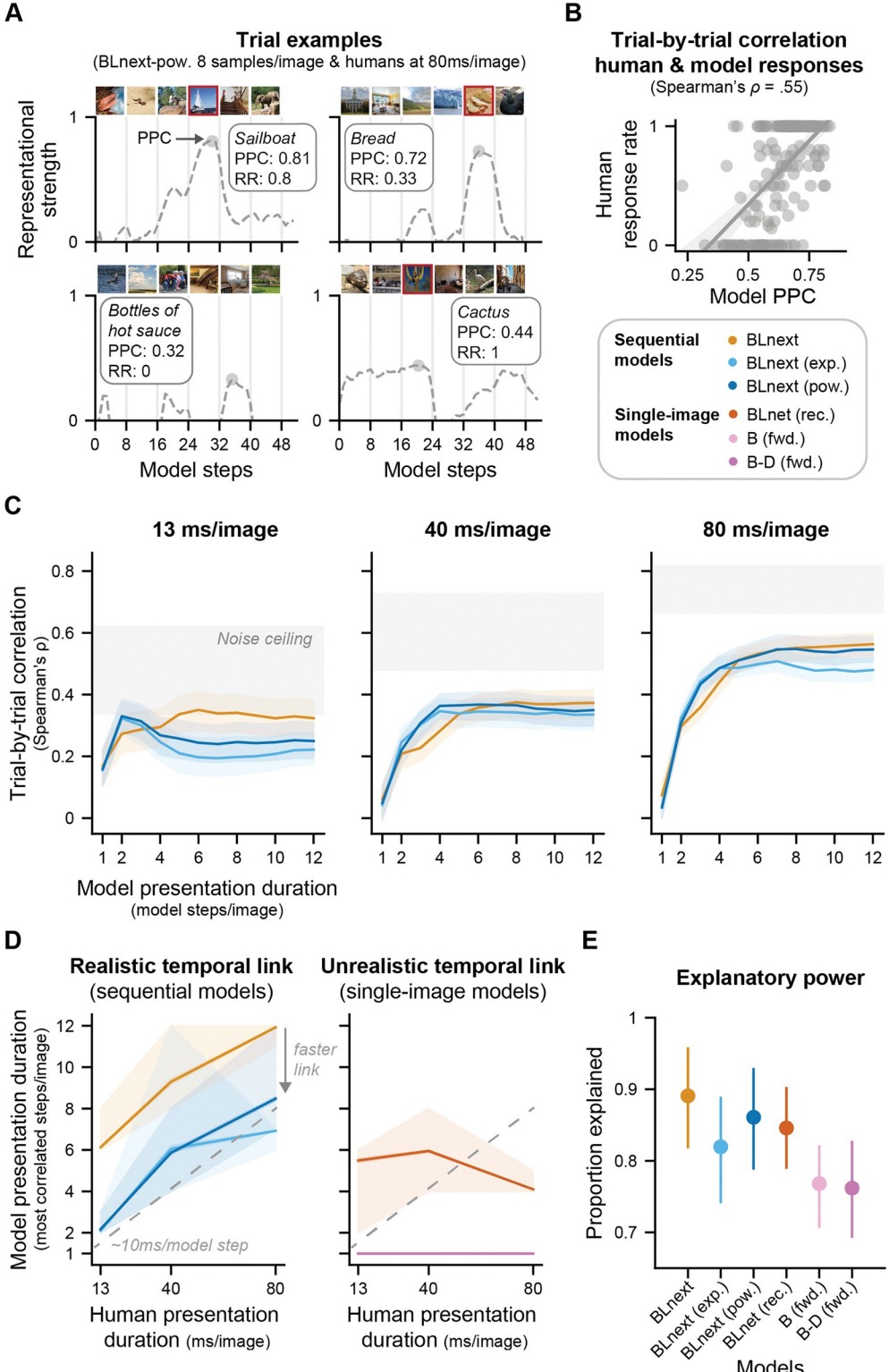

**Fig 5. Accounting for single-trial human behaviour across presentation durations reveals temporal correspondence between sequential models and human observers. A** Comparing model and human responses on a trial-by-trial basis. For every unique image sequence, we can compare the average response rate (RR) for a given presentation duration (here 80 ms/image) with the sensory evidence (PPC) shown by a given model. Target images are marked by a red frame and all images are examples from the public domain, which were chosen to be similar to the

experimental images. **B** Trial-by-trial correlation between human and model responses was quantified as the Spearman correlation between a model's PPC and the average human report rate for a given set of model steps and presentation durations. Single dots represent a single trial. The line is a fitted linear regression and only serves illustrative purposes. This example comparison shows human response rates at 80ms/image and *BLnext* (pow.) model PPCs for 8 steps/image. **C** Trial-by-trial correlations shown for all human presentation durations and sequential models (for different model presentation durations). Higher correlation values denote that the model output shown for different model steps per image can better capture the single trial behaviour shown by humans at a given presentation durations (different panels). The shaded areas indicate the 95% confidence intervals bootstrapped across subjects for all panels. The noise ceiling indicates the range of the highest possible correlations given the variability in the participants behaviour. Trial-by-trial correlations for the single-image models are shown in S3 Fig. **D** Only sequential models show a realistic temporal correspondence with human observers. To assess the temporal link between a model's and human responses, we identified the model steps per image with the highest correlation (repeated for different bootstraps) for a given human presentation duration (panels in C). A positive monotonic link indicates a realistic temporal correspondence between human and model time (left panel), whereas a flat or negative relationship indicates a mismatch in temporal correspondence (right panel). In particular, sequential models show a positive linear relationship, which is absent for single-image models. Models with adaptation use fewer model steps compared to models without adaptation, suggestive of a faster temporal link. Feedforward models only featured a single time-step and are overlapping. **E** Overall explanatory power across models is quantified as the proportion of the lower noise ceiling that is reached by the model's correlation. This estimate was averaged across presentation durations. *BLnext* models are on par with *BLnet* suggesting that while *BLnet* has superior task performance, it does not better explain variance in human behaviour across trials.

presenting it for more model steps. Such a pattern is absent for both recurrent models without adaptation (*BLnext* in Figs 5C and *BLnet* in S3). For longer presentation durations (40 and 80ms/image), we can see a more gradual increase in predictivity for the recurrent models, which is absent for feedforward models due to their lack of temporal dynamics.

As explained above, a strong test for any model of human dynamic object recognition is to assess whether it can capture how performance changes over presentation durations. To this end, we next identified for every human presentation duration, the model presentation duration of every model displaying the highest correlation (Fig 5D). A realistic temporal link between model and human presentation duration should be monotonically increasing. Put differently, a model's responses with few model steps per image should be most correlated to human responses at the fastest presentation durations. Conversely, a model with many model steps per image should be most correlated to participant responses at slow presentation durations. Our analysis revealed that only sequential, recurrent models showed this expected positive (i.e., monotonically increasing) relationship (see Fig 5D, left panel). In contrast, single-image models either were not designed to examine this (B, B-D, both feedforward models) or showed an irregular negative relationship (*BLnet*, recurrent, Fig 5D, right panel). Together, these findings indicate a selective temporal correspondence between sequential model and human performance, in line with the notion that sequential integration is a central mechanism underlying improvements in object recognition with longer presentation durations shown by human observers. Intriguingly, we also see that power-law adaptation sped up this temporal correspondence while still maintaining a linear relationship to presentation duration (Fig 5D, left panel). While for a sequential model without adaptation, a 13 ms/image presentation duration corresponded to the processing of an image for six model steps, for a model with adaptation, this was achieved in only two model steps. This latter finding implies that given a limited number of samples and model steps for every image, models with adaptation faster account for human detection performance at ultra-rapid presentation rates. In other words, adaptation is an effective means to accelerate a model's representational dynamics.

Finally, we assessed the overall explanatory power of all candidate models. It could be that a model can capture the temporal correspondence between model and human presentation durations (Fig 5D) but that this is achieved at the expense of its explanatory power. To evaluate this possibility, we calculated the proportion of the lower noise ceiling that was explained by

the best model presentation duration for every presentation duration and averaged these proportions across all presentation durations, separately per model (Fig 5E). The best possible model can take a value of 1, fully explaining the shared variance between subjects. This analysis revealed a clear pattern: All recurrent models had greater explanatory power, better predicting the single-trial data across presentation durations, than the feedforward models. This is interesting because the *B-D* model contains the same number of parameters and is thus, in principle, as expressive as a recurrent model. Furthermore, we can see that *BLnext* and *BLnext* with power-law adaptation are on par with *BLnet* in terms of their explanatory power. This finding may seem surprising at first since the performance and sensitivity results (Figs 3 and 4) indicated that *BLnet* holds better representations to solve RSVP tasks, reflected in its high accuracy and perceptual sensitivity. So, one may have expected that *BLnet* would also better explain human behaviour. Instead, we see that adding sequential processing did not hamper the model's explanatory power of human single-trial target reports, in contrast to its effects on average performance. Yet, *BLnext* also better tracked human average RSVP performance, while *BLnet* showed outperformed humans with one single model step (Fig 4). Both sets of findings thus indicate better correspondence in dynamic object recognition between sequential models (*BLnext*) and human observers.

To summarize, investigating the trial-by-trial representational dynamics revealed a number of key features of our examined computational models. First, temporal correspondence, a mapping between model and human presentation durations using trial-by-trial responses, is a hallmark of sequential models (*BLnext)* that does not come at the expense of overall explanatory power. Furthermore, adaptation is an effective mechanism to speed up this temporal correspondence, providing a model with the same representational quality after only a fraction of model steps compared to a model without adaptation. Taken together, given the rapid nature of visual stimulation in RSVP tasks, this makes a sequential, recurrent model with power-law adaptation the most plausible model of the neural computations that underlie object recognition under rapidly changing input conditions.

## Discussion

In this study, we investigated the mechanisms enabling dynamic object recognition: our ability to recognize objects across varying timescales and within a glance [2,4,8] in a constant stream of visual change. While primate object recognition is increasingly well explained by modern deep learning models optimized for computer vision for single images presented in isolation, they intrinsically cannot well capture object recognition in dynamic contexts and across varying timescales. Therefore, here, we developed a class of models capable of sequential object recognition. We used these models as a testbed to investigate the functional contributions of different neurally plausible computational mechanisms (e.g., [47,48])–lateral recurrence and sensory adaptation. There were four main findings. Our first main finding was that both lateral recurrence and sensory adaptation clearly boosted sequential object recognition performance (Fig 3D). That is, whereas sequential recognition performance was poor compared to single-image recognition performance (an easier task serving us an index of overall image difficulty) based on feedforward processing alone, this gap in performance was effectively reduced by giving models more opportunity for lateral recurrent processing and sensory adaptation. This result highlights how both mechanisms can actively support object recognition in dynamic settings, a common scenario for biological visual systems. In a next step, we compared these models to human behaviour on a second dynamic recognition task. Specifically, we quantified the effects of feedforward versus recurrent processing, single versus sequential image processing, and adaptation versus no adaptation on performance on a cued RSVP task and linked these

effects to human behaviour across different presentation durations (13–160 ms/image). Our second main finding was that only models that integrate images sequentially via lateral recurrence mirrored human performance across different image durations (13–80 ms/image). That is, while object recognition for this task was possible based on a single model step (equivalent to our definition of the neural feedforward sweep, Fig 4), sequential processing combined with lateral recurrence brought about a further increase in perceptual sensitivity with increasing image presentation durations as seen in humans. This finding supports the efficacy of the first feedforward sweep for core object recognition [2,5,7,8] in this kind of task, but also underlines that lateral recurrence, at least at the computational level, contributed to dynamic object recognition at a glance. Our third main finding was that a sequential model also best predicted human target reports on a trial-by-trial level and displayed a temporal correspondence with human observers (Fig 5). Interestingly, the most explanatory models always included some recurrence, showing that recurrent integration, in this computational framework, is a key factor for explaining trial-by-trial reports (including hits and false alarms), even at the most rapid presentation rates. Finally, our fourth main finding was that augmenting a sequential model with adaptation, and in particular, power-law adaptation, accelerated a model's representational dynamics, such that human trial-by-trial reports could be accounted for using fewer model steps (Fig 5). This suggests that adaptation can render a model's representational dynamics more efficient and can enable an observer to respond to changes more swiftly. Together with our first main finding, this indicates a key role for adaptation in dynamic object recognition by effectively increasing the quality of representations in dynamic image sequences. This corroborates a large body of evidence advocating for neural adaptation as a canonical neural computation [18,20,21,49]. Our results also substantiate the notion that dynamic object recognition is not only determined by image feature processing itself, but also critically depends on segmentation of the image stream into separate visual events [23,24,27,50]. Taken together, our findings reveal how sequential processing, recurrency, and adaptation may jointly characterize dynamic object recognition in humans.

The development of task-optimized and image-computable models has revolutionized our understanding of object recognition (e.g., [46,51,52]). For example, using these models, researchers have addressed key issues regarding the relevance of feedforward and recurrent processing in object recognition [36,53–55], yet so far mostly in the context of static images. Here, we examined the role of recurrent processing for dynamic object recognition. A key debate in the RSVP literature centres on the question to what degree core object recognition [5,6] relies on recurrent processing. In line with feedforward models of perception, some findings indicate that object recognition based on only 13ms/image can be achieved solely through feedforward processing [2]. However, findings from other studies suggest that some recurrent processing is necessary [4]. We here revisited this debate from a computational modelling perspective, asking whether recurrent processing is functionally necessary. Notably, our results seem to reconcile these two sets of behavioural findings. Object recognition was possible based on a single model step (equivalent to our definition of the neural feedforward sweep, Fig 4) in the RSVP task, supporting the idea that feedforward processing can suffice for object recognition [2,5,7,8]. Yet, human single-trial report rates were best captured by a model with at least some recurrence also at the fastest presentation rate (Fig 5). Thereby, our results underline the relevance of recurrent processing for explaining the trial-by-trial idiosyncrasies, that is, correct and incorrect target reports. It has been proposed that natural images are relatively ineffective masks [4], that can vary in their masking strength due to differences in image features. Our findings support this notion by showing that the variability in the target reports was best explained by a recurrent model, able to account for the strength of masking induced be a given image sequence. Our findings also connect to a larger body of work showing the central role of

recurrent processes in neural data using RSVP paradigms [14–16]. They critically add to these findings by directly connecting feedforward and recurrent sequential processing to human task performance and behavioural patterns during dynamic object recognition within a glance.

Our computational approach also demonstrates the functional significance of (power-law) adaptation for dynamic object recognition (Fig 3). Comparing models with and without adaptation showed that models with adaptation had superior object recognition performance, perceptual sensitivity and faster representational capacity than sequential models without adaptation in a context of dynamic visual stimulation (Figs 3D, 4C, and 5C). These findings align with past empirical work that has demonstrated a critical role for adaptation in increasing the saliency of novel stimuli or novelty detection [23,24,27,50] and that has related adaptation to efficient coding strategies in sensory systems [18,20,21,49,56]. To our knowledge, our study is the first to show such advantages of adaptation in the context of a task-performing object recognition model (but see [57,58] for temporal sequence and reinforcement learning tasks). A striking aspect of the present findings was that the combination of lateral recurrence and neural adaptation gave rise to a swift recovery of representational traces after an image changed (Fig 3B and 3D). This finding points to the possibility that rapid sensory adaptation serves as a dynamic break for recurrent computations, a crucial feature for any system that has to trade off perceptual stability or accuracy (evidence accumulation) with flexibility or speed (novelty detection). In many ways, our observation supports proposals on efficient coding [21], and might reflect a form of temporal decorrelation [22], previously linked to power-law adaptation and enhanced information processing [17,18,42]. Establishing the functional significance of computational mechanisms by studying neural systems has been a challenge in the past [23]. While recent years has seen progress in this regard (e.g., [50,59–62]), many questions still remain. Adopting performance-optimized models as done here and by [28,57] provides a promising avenue to further understanding of how and in which task contexts adaptation facilitates sensory perception and performance.

Our results favour the notion that adaptation is a canonical computation in perception, as our approach generalized adaptation mechanisms introduced to deep learning models in an innovative study by Vinken and colleagues [28] to another architecture and task. Our parameters were either identical to this former study (*exponential model*) or closely matched for short time scales (*power-law model*, see Fig 3A). As in this former study, adaptation was also introduced into the network after optimization. But here, these parameters, previously fit to single neurons by [28], were implemented in a recurrent neural network without major modifications, and shown to improve dynamic object recognition performance. These observations support the idea that adaptation is a canonical computation that is not dramatically altered by learning and specific architecture, but instead functions similarly across different visual processes and areas. Our findings also add to earlier research that put forward that power-law adaptation better captures neural data than an exponential account of adaptation over longer time scales [18,40,41,44,45]. Our data suggests that this may be due to the fact that exponential adaptation exerts a too suppressive influence at long presentation durations, a problem resolved by a power-law adaptation mechanism.

Our sequential model, *BLnext* should be seen as an important first step towards building models of dynamic object recognition. While *BLnext* generally well reproduced human performance on RSVP tasks, there were also some deviations that should be addressed in future work. For one, while *BLnet*, a single image model, reached human performance levels for images presented for 160 ms/image, *BLnext* did not reach such high accuracy in this condition (Fig 4C). Sequential processing will likely always have some negative impact on image processing at fast presentation rates (unless a model is set up to not integrate over time). Yet, ultimately, a sequential model should produce human performance across all image presentation

durations, including longer durations. That this was not the case suggests that *BLnext* could not fully harness its representational power acquired during training, even though there should have been enough time to fully represent the input. It is conceivable that the changes we introduced to build *BLnext*, such as the altered batch normalization regime, may have put *BLnext* at a disadvantage over longer time scales compared to *BLnet*. Yet, a strength of our approach was that we could closely compare *BLnet* and *BLnext* to determine the importance of sequential processing that abstracted away from differences in optimization and training history. A key avenue for future studies is to determine how sequential models can be better optimized so that they can account for dynamic object recognition across a wider range of timescales. One potential approach towards achieving this is to train models with other biologically-informed recurrent circuit motifs, such as top-down recurrence. Another observation was that we here found that the model steps that produced perceptual sensitivity levels matching human performance levels (Fig 4C) were not the same as those that best accounted for single-trial reports across presentation durations (Fig 5D). For instance, the sequential model with power-law adaptation required two model steps to best explain single-trial reports at 13ms/image, resulting in a performance of a d' of 1, which was closer to human performance levels at 40ms/image than at 13ms/image (d' of around 0.5). This discrepancy may be explained by the fact that these are different measures: while d' summarizes across trials, we evaluated single-trial predictivity as correlation between PPC values and single-trial reports, thus abstracting away from mean performance levels.

Our study illustrates the importance of conceptually dissociating between feedforward *architectures* and feedforward *processing*. We observed that feedforward *architectures* vastly outperformed humans at 13ms/image, whereas feedforward *processing* (i.e., the initial step in our recurrent network) aligned with human performance. This finding showcases that a feedforward *architecture* can still encapsulate computations that go beyond the first feedforward sweep of information processing in the brain, and is in line with the observation that some feedforward architectures are equivalent to unfolded recurrent architectures under specific circumstances, such as weight-sharing across time [63]. Yet, it also illustrates that establishing the success of a computational architecture in explaining neural processing and behavioural performance may not be sufficient to draw strong conclusions about the underlying neural process and its architecture. In our study, we took a different approach, and studied recurrent neural processes across different feedforward/recurrent interaction ratios using the RSVP paradigm with different presentation rates. Critically, this approach provides stronger constraints for matching recurrent models to human behaviour. That is, we could not only evaluate a model's explanatory power when object recognition was fully accomplished, but also when image processing was interrupted by subsequent images in the stream. Over the past few decades, RSVP paradigms have been a powerful tool to delineate how visual and cognitive processes unfold and interact over time based on behaviour [1,2,12] and more recently, based on neural data [14,15,64] providing insight into the temporal dynamics of visual perception. RSVP paradigms in combination with this wealth of experimental knowledge therefore present a promising benchmark for developing more accurate recurrent computational models of the visual ventral stream.

To summarize, we investigated the computational mechanisms governing human dynamic object recognition using a newly developed model class for sequential object recognition. Using these models, we showed that recognizing objects presented in a dynamic stream is a challenging task that can be solved by sequential object recognition models, if supported by lateral recurrence and adaptation. For explaining human visual processing on an RSVP task, it was critical to model the sequential nature of object recognition, both in accounting for overall performance levels as well as for capturing trial-by-trial idiosyncrasies. In all our experiments,

we observed a key contribution of adaptation to dynamic object recognition. Taken together, these findings shed new light on the computational mechanisms that may make object recognition so fast and effective across dramatically varying timescales in a dynamic visual world.

## Methods

### Computational models

**Sequential object recognition models (*BLnext*).**   We developed a sequential object recognition network, *BLnext*, based on *BLnet* [33]. *BLnet* consists of feedforward and lateral recurrent weights and has been shown to perform competitively on two large-scale image datasets as well as to be able to predict human reaction times in an animal detection task. For our purposes, this network has the crucial advantage that it learns for most parts weights that can be iteratively applied to any number of time steps. The only weights that are time-step specific are those for the batch normalization parameters. In a set of initial experiments, however, we discovered that the network was able to perform sustainably when simply using a fixed set of batch normalization statistics (time step: 4) by default. For reading out responses from the network, we furthermore replaced the softmax with a linear activation function. These two changes made it possible for the network to perform on a vastly longer time scale (we assessed $t > 100$ experimentally, corresponding to the 12-fold increase in model steps). In contrast to a softmax readout (with its exponential operation), a linear readout does not rescale the output distribution and skews it towards its maximum, and we found that this was helpful for obtaining smooth representational traces over model steps (see S1 Fig) and for working with the prototypes (see below) and image-level representations. We also followed this approach for evaluating our models' top5-accuracy for every model step. Specifically, a model's response was correct if the presented image at a given time step was in the top 5 of predicted output categories in the output for that model step. We chose the top 5 accuracy to account for the lingering representations of previously presented images. For all sequential models, we adopted the pretrained weights optimized on EcoSet [38] reasoning that the 565-dimensional output space is more suitable for comparisons with human performance on an RSVP task compared to an output space from ImageNet.

**Single-image object recognition models.**   We compared several single-image recognition models to our *BLnext* variants performing sequential recognition (Table 1). The task solved by single-image model was easier than that solved by sequential models, since a single-image model does not have to deal with the transitions between images as it processes every image in isolation. All single-image models stem from [33] and are thus directly comparable with regard to their architecture and optimization. Specifically, we evaluated a recurrent model (*BLnet*), a feedforward network matched to a single feedforward pass in the recurrent models (*B*), and a feedforward model with the same number of parameters as the recurrent models (*B-D*). All models weights were retrieved from https://osf.io/mz9hw/ and trained on EcoSet [38]. As for the sequential models, we also replaced the last softmax activation function with a linear function.

**Categorical prototypes.**   To compare model performance to human performance on an RSVP task, we had to develop an approach to extract a response that captures the match between the indicated category cue on a given trial and the shown image sequences. Commonly, this is achieved by retraining an output layer of a DCNN [54], but in our case, where every trial contained a unique target category, this was not feasible, nor desirable, since finetuning an output layer for every trial also introduces training history effects, thus creating differences between trials.

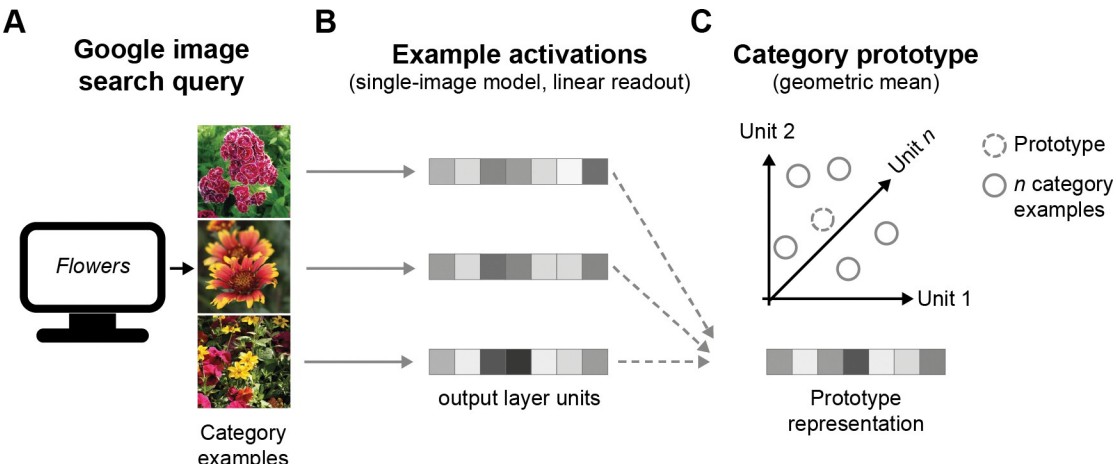

**Fig 6. Three-step approach to obtaining prototype representations. A** A large-scale image search query provided cue category examples. For every cue category (i.e., trial), we performed a separate image search query returning multiple hundreds of photographs specific to every category cue seen by the human participants. **B** Describing categories in model representations. In a next step, we took all retrieved images and presented them to the single-image models (also serving as a baseline for the sequential models). Like this, we obtained an activation pattern for every image. **C** Computing the prototype representation. Together all category examples formed a category manifold. To obtain a prototype representation, we simply found the geometric mean across the activation patterns evoked by the different category exemplars. Note that only a subset of exemplars was used to obtain the final prototype.

In many ways, this challenge resembles few-shot learning as discussed by Sorscher and colleagues [65] in which the pre-trained representational space is leveraged to perform untrained, categorical distinctions (also see [66]). In this spirit, we here also adopted categorical prototypes, which were obtained by estimating the geometric centre from activations belonging to different category examples. We obtained these examples using google image search. Specifically, we used the trial target categories as search queries (see Fig 6A). We then filtered these queries based on a set of criteria to increase the usability of the retrieved images (images being a full-colour photograph, in.jpg format and distributed under a 'labeled-for-noncommercial-reuse-with-modification' license). This approach provided a large set of images (*N*) for every category cue (average number of images per category: 367, min–max: 262–396). In a next step, we presented these images to the single-image models and retained their activations (Fig 6B). Note that *BLnet* and *BLnext* share their weights and consequently, categorical prototypes for *BLnext* were identical with *BLnet*. In principle, these activations form the categorial representations linked to a category cue, which allowed us to compute their geometric centre to obtain the prototype representations (Fig 6C).

We did not use the activations of all category exemplars, but rather, selected some exemplars based on their semantic relevance. We took this step because during initial quality checks of these obtained images, it became apparent that some categories suffered from substantial semantic ambiguity. For instance, the obtained images for the category 'Ranger' contained images from both the football team, the profession, as well as the car type. To prevent such ambiguity effects from confounding our findings, we used the average representation evoked by a target and a foil image as a marker for semantic relevance. These foil images were chosen to be congruent with the category cue in the original study and were presented as a 2-AFC target identification task directly after the detection task to test whether participants could identify the target images. We reasoned that whatever was shared between the target and foil image would be a good approximation of the intended category. Using this notion of semantic relevance, we established a ranking of the most semantically relevant exemplars. Based on this

relevance ranking, we calculated prototypes including the activations of either the 10, 50 or 100 most relevant exemplars. For all experiments, we report results for a prototype based on the 10 most relevant exemplars. Task performance results for prototypes computed based on more exemplars were very comparable and mainly scaled the observed pattern of results (S4 Fig).

For the sequential models, we estimated the representational strength $p$ at time $t$ for a given prototype $\phi$ and model output $y$ by computing the Pearson correlation coefficient $r$:

$$p_t = r(\phi, y_t),$$

providing a measure of how well the current model state $y_t$ approximates the prototypical activation $\phi$. For single-image models, the same correlation was computed between the response to every image (processed in isolation) in the sequence and the prototypical activation $r$. For *BLnet*, the correlation was computed after the specified number of model steps.

**Adaptation mechanisms.** We implemented neural adaptation as activation-based intrinsic suppression [28], targeting all activation functions in a sequential model (*BLnext*). In particular, this approach introduced an intrinsic adaptation state $s_t$ that increased or decreased as a function of the recent sensory history of a unit. This term was scaled by a constant $\beta$ and then subtracted from the incoming activation to a unit ($b + Wx_t$) before the non-linearity $\sigma$ is applied, resulting in $a_t$:

$$a_t = \sigma(b + Wx_t - \beta s_t)$$

Depending on $\alpha$, $s_t$ either quickly accumulates due to recent activation $a_{t-1}$ or decays its suppression over time following an exponential function:

$$s_t = \alpha_e s_{t-1} + (1 - \alpha_e)a_{t-1}$$

Following [28], we here chose $\alpha_e = 0.96$ and $\beta = 0.7$ and a ReLU as non-linearity $\sigma$.

The implementation of a power-law adaptation mechanism is identical to the formulation above, with only the intrinsic adaptation state being different:

$$s_t = \sum_{\alpha_p}(\alpha_p s_{t-1} + (1 - \alpha_p)a_{t-1})$$

Based on [40], we approximated a power-law adaption state as a sum of exponentials with $\alpha_p = [0.96, 0.75]$ and $\beta = 0.15$. Note that $\beta$ is now much lower to accommodate the larger sums in $s_t$. We chose the second exponential such that it would match the exponential adaptation mechanisms on short timescales but would be less suppressive over longer time scales.

## Behavioural study in humans

The human behavioural data used in this study was collected in the context of another study that investigated the role of mid-level visual features and altered category information on object recognition during RSVP. This behavioural study also included an exact replication condition of [2], including the same stimuli and presentation sequences (only the presentation rates were different in [2] with 13, 27, 53, or 80 ms per image). For the current study, only the data from this condition was used to evaluate model performance. For more information on the aims and hypotheses of this behavioural study, please see https://osf.io/b7fdq/.

**Ethics statement.** Behavioural data collection was carried out in accordance with the recommendations of the ethical committee of the University of Amsterdam (approved as 2018-BC-8763). All participants gave written informed consent before study onset in

accordance with the Declaration of Helsinki and were paid €10 per hour or participated for research credit.

**Participants.** 36 participants with an average age of 22.43 years (18–29, SD = 2.98, one participant did not indicate an age) participated in the behavioural study. All participants had normal or corrected-to-normal vision, were not colour-blind, between 18 and 35 years old, had no history of psychiatric or neurological disorder, nor a personal or family history of epilepsy.

**Task.** On every trial, participants viewed a sequence of six real-world photographs, each presented for either 13, 40, 80 or 160 ms per image, and were tasked to determine whether an image of a target category was present in the sequence (see Fig 7A for a schematic). This target category was a conceptual description of the image (e.g., flowers) and was communicated to the participants prior to the image sequence, and repeated when they had to give their response. Participants initiated a trial with a button press and indicated their answer with a button press ("v" for yes, "b" for no). They received feedback on their response, by either seeing the text "No target" (when no target was shown) or, in case there has been a target image, being asked to identify the image they saw from a set of two images. A quarter of all trials did not contain a target image. Target images could never be the first of last image in the sequence. For the presentation durations of all trial elements, please see Fig 7.

The experiment consisted of a practice block with a slow presentation rate (133 ms/image), which was followed by eight experimental blocks, each featuring 22 trials. After every block, participants had the opportunity to take a break. Every image sequence was shown once to every participant, such that every trial had unique images and a unique target category. In experimental blocks, all trials were randomized with regard to image presentation duration with the purpose of showing every sequence equally often across presentation duration across participants.

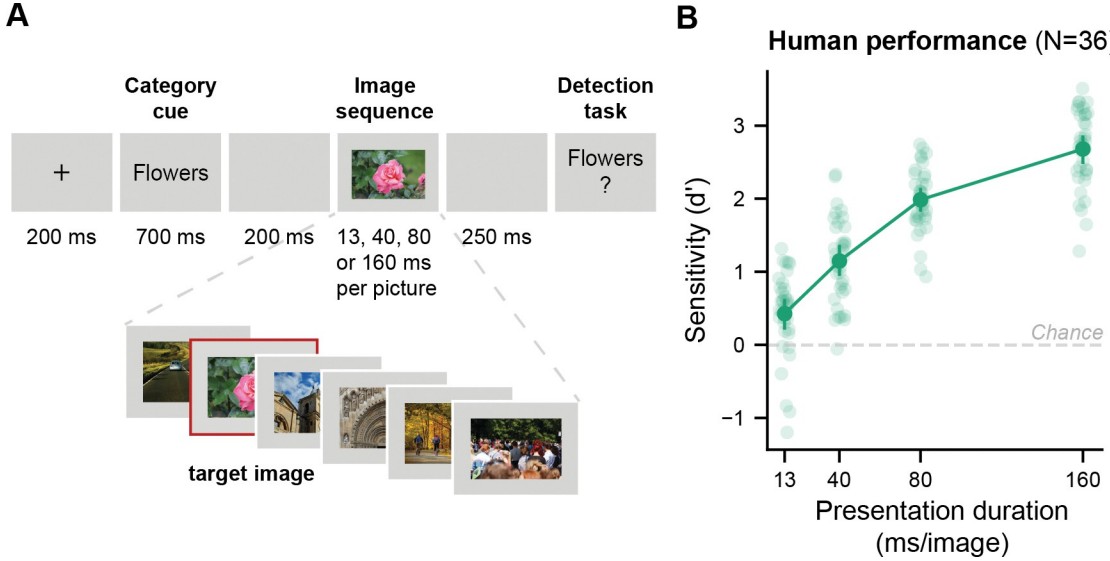

**Fig 7. The RSVP task and human performance across different presentation durations. A** Schematic of a trial in the RSVP paradigm used in the behavioural study in humans. We used the same image sequences and category cues as in a study by Potter et al., (2014) [2] but varying the presentation durations. Participants started a trial with a button press after which a category cue was shown. Participants were tasked to determine whether an image matching this category cue was present in a subsequent stream of six real-world photographs, presented at a varying duration (13, 40, 80 or 160 ms per picture). At the end of every trial, participants indicated whether an image matching the category cue was present in the sequence or not. **B** Perceptual sensitivity of human observers across different presentation durations. Replicating Potter et al. (2014), participants performed above chance for all presentation durations, including at 13ms/image (all *p's* < .0001). The shaded dots represent individual participants, error bars show the 95% confidence interval for the mean in d' across participants.

**Apparatus.** The stimuli were presented on a CRT screen (refresh rate: 75Hz; resolution: 1.204 x 768) and stimulus presentation was controlled via Psychophysics Toolbox Version 3 [67,68] implemented in MATLAB. Presentation timing was continuously monitored during the experiment and also measured externally using a photo-diode prior to data collection. No trials were excluded based on timing errors of the presentation duration of the target image ($> \pm 12$ms).

**Analysis.** Perceptual sensitivity (d') was calculated for each presentation duration and participant (see *Perceptual sensitivity in humans and models*). For each presentation duration separately, a planned two-tailed *t*-test was conducted to determine whether perceptual sensitivity differed from chance performance (i.e., *d'* of 0).

## Perceptual sensitivity in humans and models

Task performance on the RSVP task was assessed in d', a measure for perceptual sensitivity from signal detection theory that abstracts away from the perceptual criterion subjects use. For human observers and model responses, the true positive rate *TPR* was described across all trials by

$$TPR = \frac{TP}{TP + FN}$$

with *TP* referring to correct responses on trials with a target and *FN* to incorrect responses on target trials. Analogously, the false positive rate *FPR* was defined by

$$FPR = \frac{FP}{FP + TN}$$

with *FP* being incorrect responses on target absent trials and *TN* correct responses on target absent trials. Note that both *TPR* and *FPR* depend on the observer's criterion *c*, which is set internally by a human observer. For human observers, we followed the log-linear method [69] by adding 0.5 to every trial type correcting for extreme values, such that $d'_{human}$ is defined as:

$$d'_{human} = z(TPR_{loglinear}) - z(FPR_{loglinear})$$

with *z* as the z-score, thus the inverse of the normal cumulative distribution function.

In contrast to human observers, the model does not produce a response but rather we can gather a sensory evidence time course *p*. To obtain a response to a given trial, we define the peak prototype correlation *PPC* for a given trial:

$$PPC = max(p)$$

For single-image models, the *PPC* was defined as the maximum across all presented images instead of model steps.

Using the *PPC*s across all trials we can then determine the area under curve (*AUC*) for the receiver operating characteristic (*ROC*):

$$AUC = \int_{c=0}^{1} \text{TPR}(\text{FPR}^{-1}(c)) dc$$

with *c* being the range of possible criterion values applied to trial-by-trial *PPC*s, thereby resulting in different TPR and FPR, respectively. Finally, this *AUC* can be translated into *d'* by

$$d'_{model} = \sqrt{2} z(AUC)$$

### Single-trial prediction

**Predictivity.** We determined the predictivity $\rho$ of a given model and model step duration based on the Spearman correlation between the PPC across all trials and the average response rate ($RR$) across participants for a given presentation duration and image:

$$RR = \frac{P}{N}$$

$$\rho = r_s(PPC, RR)$$

$RR$ captured the probability of participants to report a target, irrespective of whether the image sequence indeed contained a target image or not, where $P$ refers to the number of yes responses across subjects and $n$ to the number of subjects. We chose here to model the report rate rather than the hit rate to be able to include all trials in our estimate of $\rho$. We estimated $\rho$ repeatedly based on 1000 bootstraps drawn across subjects to estimate the 95% confidence intervals for all results presented in Fig 5.

**Noise ceiling.** To understand how well any model could possibly predict the participant's responses, we estimated the upper and lower noise ceiling. For constructing the upper noise ceiling, we computed the Spearman correlation between every individual participant's single-trial report with the group average report ($N$). The final upper noise ceiling was then the average of this correlation across all subjects. The lower noise ceiling was obtained following the same procedure, but without the individual single-trial report that is currently compared contributing to the group average report ($N$-1).

**Temporal correspondence & explanatory power.** To investigate the correspondence between presentation durations and model steps, we identified the most predictive model step for every model and image presentation duration. To estimate the overall explanatory power of a given model, we first converted the most predictive model step's $\rho$ into a proportion of the lower noise ceiling, thereby normalizing correlations across different presentation durations. After doing this separately for all presentation durations, we simply took the average to obtain the overall explanatory power. Both the temporal correspondence and the explanatory power were determined for every bootstrap of subjects and visualized in the 95% confidence intervals in Fig 5.

### Software

In addition to custom code, the results presented here were obtained while relying on the following Python packages: NumPy [70], keras [71], TensorFlow [72], Pandas [73], Pingouin [74], and SciPy [75]. Data visualization was done using matplotlib [76] and, in particular, seaborn [77].

### Supporting information

**S1 Fig. Comparison of a linear and softmax readout for categorical traces.** *BLnet* was optimized with a softmax readout using categorical labels. A softmax operation applies an exponential function to all elements and normalizes them to a sum of 1. This leads to a distribution in which the highest value dominates and as result such a transformation is useful for matching a categorical vector. This principle is illustrated here in the correlation values for a linear and softmax readout. While a linear readout (no skewing toward the maximum value) produced a rather low correlation with the categorical vector (left panel). A softmax readout produces much more extreme correlation values due to its winner-take-all properties,

renormalizing across all target classes (right panel).
(TIF)

**S2 Fig. Adaptation mechanisms do not benefit single-image recognition.**
Figure conventions follow those in Fig 3D. In contrast to a sequential recurrent model
(*BLnext*), a single-image recurrent model (*BLnet*) does not improve its performance after
including adaptation mechanisms (i.e., exponential or power-law).
(TIF)

**S3 Fig. Trial-by-trial correlations between single-image models and human responses
across different presentation durations.** Figure conventions follow those in Fig 5C.
(TIF)

**S4 Fig. Varying the number of prototype exemplars scales baseline sensitivity levels, with-
out changing the pattern of results.** Using prototypes that are based on more exemplar
images leads to a reduction in performance levels that affects all model types in a similar fash-
ion. Note that performance is reduced as a result of increasing the number of images because
the exemplar images have been ranked with regard to their distance to the mean of the target
and foil image. This step was necessary to ensure that the prototype was informative and
unambiguous (see *Methods*). Without such ranking, one would expect performance to increase
as a function of exemplars.
(TIF)

## Acknowledgments

The authors would like to express their gratitude to Mary C. Potter and Carl Hagman who
made this study possible by sharing their stimuli and experimental design with us. We also
would like to thank Courtney Spoerer and his co-authors for making *BLnet* and its control
models openly available.

## Author Contributions

**Conceptualization:** Sander M. Bohté, Heleen A. Slagter, H. Steven Scholte.

**Data curation:** Lynn K. A. Sörensen.

**Formal analysis:** Lynn K. A. Sörensen.

**Funding acquisition:** Sander M. Bohté, Heleen A. Slagter, H. Steven Scholte.

**Investigation:** Lynn K. A. Sörensen, Dorina de Jong.

**Methodology:** Lynn K. A. Sörensen, Dorina de Jong.

**Project administration:** Lynn K. A. Sörensen, H. Steven Scholte.

**Resources:** Lynn K. A. Sörensen, H. Steven Scholte.

**Software:** Lynn K. A. Sörensen, Dorina de Jong.

**Supervision:** Sander M. Bohté, Heleen A. Slagter.

**Visualization:** Lynn K. A. Sörensen.

**Writing – original draft:** Lynn K. A. Sörensen.

**Writing – review & editing:** Lynn K. A. Sörensen, Sander M. Bohté, Heleen A. Slagter, H.
    Steven Scholte.

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
