## [Decision Letter · Decision Letter 0]

14 Mar 2023

Dear Mrs. Sörensen,

Thank you very much for submitting your manuscript "Mechanisms of human dynamic object recognition revealed by sequential deep neural networks" for consideration at PLOS Computational Biology. As with all papers reviewed by the journal, your manuscript was reviewed by members of the editorial board and by several independent reviewers. The reviewers appreciated the attention to an important topic. Based on the reviews, we are likely to accept this manuscript for publication, providing that you modify the manuscript according to the review recommendations. Please clarify the availability of any data used in your manuscript. 

Sincerely,

Matthieu Louis

Academic Editor

PLOS Computational Biology

Marieke van Vugt

Section Editor

PLOS Computational Biology

Reviewer's Responses to Questions

**Comments to the Authors:**

Reviewer #1: The authors present a thorough analysis and potential explanations for how the human visual system processes dynamic input, a topic that is currently receiving much attention. The authors made their data and models available which allowed me to reproduce the results and see the code. This is an excellent paper, well written and very interesting, and I only have minor suggestions.

When introducing the BLnet on p5, the authors should justify the rationale for choosing BLnet over any other DNNs.

In the next paragraph, BLnext is mentioned but it has not been explained yet what BLnet is and how it is a modification of BLnet

In Table 1, one may wonder why there is no option for a model that uses dynamic input but without lateral recurrence

P4 adaption->adaptation

Reviewer #2: In this paper, Sörensen and colleagues study dynamic object recognition in recurrent convolutional neural networks, some of which were augmented with neural adaptation mechanisms. Feedforward convolutional neural networks (CNN) have become powerful tools to model ventral stream processing, but they lack the temporal dynamics of our visual system. Previous recurrent CNN models have temporal dynamics, but those studies focused on the “depth through recurrence” principle for static input. In contrast, Sörensen and colleagues studied this class of models with more natural, dynamic input. They show that, like in humans, recurrence hinders processing under highly dynamic conditions, but that some of that impediment is rescued by a neuronally intrinsic adaptation mechanism known to be prevalent in sensory neurons.

I thought this was a carefully conducted computational study that touches upon many subdomains of the field by bridging the gap between neural mechanisms and complex behavior. I greatly appreciate how the authors combined dynamic processes from previous models, and challenged the resulting model with a new task in order to further our understanding of the role of these processes. This led not only to an improved model of dynamic ventral stream processing, but also to a novel perspective on adaptation, namely its potential role in modulating _recurrent_ processing under dynamic conditions. This work neatly shows how complex, task-performing neural networks are an important tool to shed light on neural mechanisms and how they might interact. Specifically for adaptation, up until now the functional significance of this mechanism has been subject to mostly speculation.

Overall I felt like the paper was well written and clear, and the conclusions are well supported. I do have some comments that I would like to see addressed or that need clarification for me. I will discuss these in more detail below.

MAJOR

One major question that I had which was left unaddressed, is how adaptation affects the performance of a recurrent CNN without preceding stimulus. That is, if only one stimulus is presented (e.g. like in the regime where BLnet was trained, or if the target is the first stimulus of a sequence), does adaptation: (a) negatively impact performance, (b) have no real effect, or (c) help?

Answering this question would help to understand whether adaptation specifically helps in a fast-changing environment, or whether it just generally impairs recurrence. If it (a) generally impairs recurrence, adaptation may be helpful in a dynamic context, but could be hurtful when the recurrence is needed in a more static context. Alternatively, (b) it could be that adaptation helps specifically in a dynamic context because it prioritizes feedforward input over recurrence only after the input has changed drastically. In that case there might be no real effect in situations where the input is more constant. A third possibility (c) is that adaptation always renders recurrence more efficient – also when there is no preceding stimulus – by for example dampening redundant feedforward signals.

MINOR

- I was curious how a feedforward model with adaptation would perform. Adaptation by itself could increase performance by highlighting target-specific information, but it could also have a detrimental effect.

- Abstract: What result does this sentence refer to (specifically: it sounds like it refers to the same results as the previous sentence): “Importantly, sequential lateral-recurrent integration was also necessary for a model to display a plausible temporal correspondence with human performance”.

- Fig 2, legend: “representational strength can be computed … as a distance” -> I presume not distance, but similarity/correlation?

- P10: “We evaluated all sequential models on 500 randomly generated images sequences” -> It would be useful to specify whether the target could be placed at all 6 positions.

- Fig 5B: Is this for all stimulus durations or for 80 ms only?

- P20, Methods, Categorical prototypes: What is a “foil” image?

- P20, Methods, Categorical prototypes: “These images stem from the 2-AFC task…” -> does “these images” refer to the foil images of the previous sentence?

- P24, Methods, Single-trial prediction, Predictivity: I was confused by the equation for “RR”. I thought RR was calculated for each specific trial, but a target cannot be both absent and present in the same trial. In other words, if a target is present, FP and TN don't apply, and if a target is absent TP and FN don't apply. Or am I missing something?

Reviewer #3: The article contrasts human performance in RSVP tasks (with rapidly changing images) with deep neural networks trained on image recognition. While humans show performance decrements when the rate of image presentation increases, feedforward deep nets are insensitive to this manipulation by construction. The authors ask if a recurrent type of convolutional network (BLnext, an adaptation of the BLnet model from Spoerer et al) would show similar behavior as humans. And indeed, the recurrent network is affected by faster presentation speeds (less steps per image). Considering that BLnet was trained for optimal recognition over 8 time steps with a constant input image, this finding is not altogether surprising (the less time steps are allowed for each image, the more the testing conditions will depart from the training). Nonetheless, there is some amount of quantitative correspondence between the performance changes observed in humans and in the model. In addition, trial-by-trial variations in accuracy are also matched between humans (at a particular RSVP presentation speed) and the BLnext model at a fixed number of time steps per image, and this match is higher than for feedforward networks. This latter result is non-trivial in my opinion. Finally, the authors also investigate the potential effects of neural adaptation on the performance time course, and find approximately the same pattern of results, but with faster dynamics. That is, the number of time steps that best corresponds to 1 second of processing in the human visual system is lower for a recurrent model with adaptation than without. Although this is not direct proof, the authors argue (based on parsimony) that this is compatible with a computational role for adaptation in explaining the temporal dynamics of human vision.

The paper is well written, the methods appear solid and the results reliable. Overall, the authors have spun their results into a plausible story, which could make a meaningful contribution to the literature on matching human behavior and deep nets. My main comments will thus mirror my lukewarm assessment above.

1. About the main finding, that a pretrained recurrent model is affected by changing image presentation rate (just like humans), while a feedforward network isn’t, nor is a version of the recurrent model that is artificially reset between image changes: I honestly don’t know what else could have been expected here. I agree that (as far as I know) this hasn’t been shown before, and that it is therefore worth reporting. But on the other hand, I don’t see an alternative outcome that could have made sense.

2. Related to this, it could have been informative to compare different types of recurrent networks, with different connectivity patterns. For instance, Spoerer et al (2017) had contrasted the BL architecture (lateral connections) with a BT architecture (top-down connections) and a BLT version (both connection types). I think all three types of architectures, if trained on a constant input image, would be affected by changing the input to an RSVP stream. But it’s not clear to me which of the three would best match human performance, so this could have been a more interesting question. As far as I know, there is no pretrained version of BT and BLT on the ecoset dataset that could be compared against BLnet, so I understand why authors did not do this, but it could be a worthwhile consideration for future work. It certainly would have made the paper more interesting for me.

**Have the authors made all data and (if applicable) computational code underlying the findings in their manuscript fully available?**

Reviewer #1: Yes

Reviewer #2: None

Reviewer #3: **No: **I didn't see a code availability statement

PLOS authors have the option to publish the peer review history of their article (what does this mean?). If published, this will include your full peer review and any attached files.

Reviewer #1: **Yes: **Tijl Grootswagers

Reviewer #2: No

Reviewer #3: No

Figure Files:

Data Requirements:

Reproducibility:

References:

---

## [Decision Letter · Decision Letter 1]

9 May 2023

Dear Mrs. Sörensen,

We are pleased to inform you that your manuscript 'Mechanisms of human dynamic object recognition revealed by sequential deep neural networks' has been provisionally accepted for publication in PLOS Computational Biology.

Best regards,

Matthieu Louis

Academic Editor

PLOS Computational Biology

Marieke van Vugt

Section Editor

PLOS Computational Biology

Reviewer's Responses to Questions

**Comments to the Authors:**

Reviewer #1: The authors have addressed all my comments and improved the manuscript significantly in this revision.

Reviewer #2: Sörensen and colleagues fully addressed my questions and I have no remaining concerns.

I would like to add one suggestion, which is that the authors could expand a bit on the implications of the new analysis of Fig. S2 in the manuscript (either briefly in the results, or in the discussion). From the short description currently in the results I am not sure whether the implications of the analysis are as clear to the reader as they are in the author’s response to me, which was:

“Fig. S2 shows that whereas BLnet models with adaptation are indistinguishable from those without adaptation for a few model steps per image, adaptation models show degraded performance when the same image is processed longer (i.e., > 4 model steps). Yet, notably, all models (incl. those with adaptation) still show comparable maximum performance. These results fit with our account that the effects of adaptation become most apparent once the benefits of lateral recurrent processing have been reaped, that is, once a model’s recurrent processing becomes repetitive. The result in Fig. S2 also clarifies how including adaptation during sequential processing helps recognition: Once a lateral recurrent model has reached its peak performance, adaptation helps to ready a model for processing new incoming stimuli. Together, these results provide further support for the notion that lateral recurrence and adaptation mutually enhance each other.”

Reviewer #3: My initial review stated that the findings, although unsurprising, could be considered worth reporting.

This has not changed for the revised manuscript.

**Have the authors made all data and (if applicable) computational code underlying the findings in their manuscript fully available?**

Reviewer #1: None

Reviewer #2: None

Reviewer #3: None

PLOS authors have the option to publish the peer review history of their article (what does this mean?). If published, this will include your full peer review and any attached files.

Reviewer #1: No

Reviewer #2: No

Reviewer #3: No

---

## [Editor Report · Acceptance letter]

6 Jun 2023

PCOMPBIOL-D-22-01860R1 

Mechanisms of human dynamic object recognition revealed by sequential deep neural networks

Dear Dr Sörensen,

I am pleased to inform you that your manuscript has been formally accepted for publication in PLOS Computational Biology. Your manuscript is now with our production department and you will be notified of the publication date in due course.

With kind regards,

Anita Estes
